# A Method for Assessing Inference Patterns Captured by Embedding Models in Knowledge Graphs

## Abstract

Various methods embed knowledge graphs with the goal of predicting missing edges. Inference patterns are the logical relationships that occur in a graph. To make proper predictions, embedding methods must capture inference patterns. There are several theoretical analyses studying pattern-capturing capabilities. Unfortunately, these analyses are challenging and many embedding methods remain unstudied. Also, they do not quantify how accurately a pattern is captured in real-world datasets. Empirical studies have been generally not consistent, and have evaluated edges in isolation.

We present a model-agnostic method to empirically quantify how patterns are captured by trained embedding models. We collect the most plausible predictions to form a new graph, and use it to globally assess pattern-capturing capabilities. For a given pattern, we study positive and negative evidence, i.e., edges that the pattern deems correct and incorrect based on the partial completeness assumption. As far as we know, it is the first time negative evidence is analyzed. The assessment of a pattern measures the similarity of the positive and negative evidence between predictions and a ground truth, the original graph. Our findings indicate that several models effectively capture inference patterns for positive evidence. However, the performance is quite poor for negative evidence, which entails that models fail to learn the partial completeness assumption, even though they were trained using it. Finally, we identify new inference patterns that have not been studied before. Surprisingly, models generally achieve better performance in these new patterns that we introduce.

## CCS Concepts

• **General and reference** → **Evaluation**; • **Computing methodologies** → **Semantic networks**; • **Information systems** → **Data mining**.

## Keywords

Knowledge Graph Embedding, Link Prediction, Inference Patterns

**ACM Reference Format:**

Anonymous Author(s). 2024. A Method for Assessing Inference Patterns Captured by Embedding Models in Knowledge Graphs. In *Proceedings of TheWebConf'24: The ACM Web Conference (TheWebConf'24)*. ACM, New York, NY, USA, 11 pages. https://doi.org/10.1145/nnnnnnn.nnnnnnn

## 1 Introduction

Knowledge graphs connect entities of interest and serve as shared, common knowledge within organizations or communities [17]. These graphs are currently at the core of many crucial services, such as search engines, social networks and product catalogs [12, 26]. Unfortunately, knowledge graphs are typically incomplete due to their unsupervised construction, i.e., there are many missing edges between entities that are indeed related [28]. An edge is a $p(s, o)$ triple, where $s$ and $o$ are entities, and $p$ is a predicate or label.

The literature is rich in machine learning methods to embed and complete a knowledge graph [19, 39]. These embedding methods train models that predict missing triples [7]. In each method, a scoring function assigns plausibility scores to input triples. Missing triples in knowledge graphs are typically considered unknown rather than incorrect, a.k.a. the open-world assumption [17]. However, embedding methods learn based on correct (positive) and incorrect (negative) triples [39]. The partial completeness assumption (PCA) is the most popular compromise: a missing triple is negative only if a part of it is in the graph; otherwise, it is unknown [7].

It is desirable that an embedding method captures a set of inference patterns, i.e., the logical relationships that may exist in the graph at hand [1]. For example, "if word $w_1$ is a meronym (part) of word $w_2$, then $w_2$ is a holonym (whole) of $w_1$" is a symmetry pattern, i.e., $meronym(w_1, w_2) \Rightarrow holonym(w_2, w_1)$. There are theoretical studies that analyze whether some of the existing embedding methods are able to capture various patterns [1, 25, 33, 35, 37, 40, 41]. We discuss the following shortcomings:

- Except Abboud et al. [1], patterns have been studied in isolation. In practice, patterns must be learned jointly, i.e., the same model must capture several patterns at once [1].
- Jointly learned patterns are grouped by type, e.g., hyponym-hypernym ("kind of") and meronym-holonym belong to symmetry [1]. How are different types learned together?
- Theoretical studies are challenging and many are missing. Also, results are binary, i.e., whether a pattern is captured or not. We thus cannot quantify how a pattern is captured.
- Pattern-capturing analyses focus solely on positive triples. Yet negative triples are important too. Has a model learned to identify negatives according to PCA?

Empirical studies are a promising alternative to address these issues. As far as we know, only Rossi et al. [29] have empirically studied inference patterns across multiple methods and datasets. They focus on link prediction that is commonly used to compare embedding methods [36]. The evaluation protocol computes positions (ranks) of positive triples w.r.t. negative triples when sorted by plausibility score, typically based on PCA [7]. Rossi et al. [29] studied Hits@1, i.e., how many positives are ranked first. Unfortunately, Hits@1 evaluates each positive triple in isolation, which provides a prediction-level, a.k.a. local, assessment rather than

**Table 1: Inference patterns and whether embedding methods capture them individually (I) or jointly (J), where $p_1 \neq p_2 \neq p_3$. If a theoretical analysis is missing, we report the first reference that did it. Dash symbols entail no analyses were found.**

| | Hierarchy $p_1(X,Y) \Rightarrow$ $p_2(X,Y)$ | | Symmetry $p(X,Y) \Rightarrow$ $p(Y,X)$ | | Antisymmetry $p(X,Y) \Rightarrow$ $\neg p(Y,X)$ | | Inversion $p_1(X,Y) \Rightarrow$ $p_2(Y,X)$ | | Intersection $p_1(X,Y) \wedge p_2(X,Y) \Rightarrow$ $p_3(X,Y)$ | | Transitivity $p(X,Z) \wedge p(Z,Y) \Rightarrow$ $p(X,Y)$ | | Composition $p_1(X,Z) \wedge p_2(Z,Y) \Rightarrow$ $p_3(X,Y)$ | |
|---|---|---|---|---|---|---|---|---|---|---|---|---|---|---|
| | I | J | I | J | I | J | I | J | I | J | I | J | I | J |
| BoxE [1] | ✓ | ✓ | ✓ | ✓ | ✓ | ✓ | ✓ | ✓ | ✓ | ✓ | – | – | ✗ | ✗ |
| ComplEx [37] | ✓[1] | ✗[1] | ✓ | ✓[1] | ✓ | ✓[1] | ✓[35] | ✓[1] | ✗[1] | ✗[1] | ✗[33] | – | ✗[35] | ✗[1] |
| HAKE [41] | – | – | – | – | – | – | – | – | – | – | – | – | – | – |
| HolE [25] | ✓[1] | ✗[1] | – | – | – | – | – | – | – | – | – | – | – | – |
| QuatE [40] | – | – | ✓ | – | ✓ | – | ✓ | – | – | – | ✗ | – | – | – |
| RotatE [35] | ✗[1] | ✗[1] | ✓ | ✓[1] | ✓ | ✓[1] | ✓ | ✓[1] | ✓[1] | ✗[1] | ✗[33] | – | ✓[35] | ✗[1] |
| RotPro [33] | – | – | ✓ | – | ✓ | – | ✓ | – | – | – | ✓ | – | ✓ | – |
| TorusE [13] | – | – | – | – | – | – | – | – | – | – | – | – | – | – |
| TransE [7] | ✗[1] | ✗[1] | ✗[35] | ✗[1] | ✓[35] | ✓[1] | ✓[35] | ✗[1] | ✓[1] | ✗[1] | ✗[33] | – | ✓[35] | ✗[1] |

global. A global assessment is desirable to understand the global relationships a model has learned [24].

In this paper, we propose an approach to empirically analyze how models capture inference patterns. First, we mine inference patterns from the whole graph at hand, which we refer to as core patterns. We measure support and confidence, i.e., the triples that a pattern deems positive and negative, respectively, which we refer to as positive and negative evidence. Second, we train a model using a training set (a subgraph of the graph at hand), and evaluate link prediction over a test set (another subgraph with no overlap) using the protocol described above. We make the following observation: predicted triples ranked in the first $k$ positions are considered more plausible with high confidence by the model under evaluation. We collect these top-$k$ predicted triples and create a new graph. We study positive and negative evidence for each core pattern in this new graph. We consider the assessment made in the first step as the ground truth, and compare how positive and negative evidence deviates. The salient features of our approach are as follows:

- Since we focus on link prediction evaluation, it is agnostic to the inner workings of the embedding methods. This allows us to homogeneously compare methods side by side.
- Rather than isolated predicted triples, it collectively considers several predicted triples made by a model. Our output is thus a global assessment of such a model.
- It quantifies how patterns are captured in terms of both positives and negatives. To the best of our knowledge, negatives have never been measured before.
- It supports multiple types of patterns, as long as positive and negative evidence can be quantified. We analyze new patterns not studied before, as far as we know.

Our experiments include a variety of embedding methods, such as BoxE [1], HAKE [41], QuatE [40] and TransE [7], and datasets, such as WN18, WN18RR and YAGO3-10. In our results, we observe that redundancy in a dataset makes models improve performance, that inverse triples impact a number of inference patterns, that models fail to identify negatives based on PCA even though they were trained under the assumption, and that models achieve better performance in the new patterns we introduce.

The rest of the paper is organized as follows: Section 2 discusses the related work; Section 3 introduces necessary background; Section 4 presents our approach; Section 5 presents our experiments; and Section 6 discusses our conclusions and future work.

## 2 Related work

Section 2.1 presents theoretical analyses conducted for several embedding methods, and Section 2.2 focuses on empirical studies.

### 2.1 Inference patterns

Table 1 summarizes the theoretical analyses of embedding methods capturing inference patterns found in the literature. An embedding method capturing an inference pattern means that, if we represent the pattern using the scoring function, it does not lead to a contradiction. For instance, RotatE captures symmetry: the scoring function is $\mathbf{s} \circ \mathbf{p} = \mathbf{o}$, where $\mathbf{s}$, $\mathbf{p}$ and $\mathbf{o}$ are vectors and $\circ$ is the Hadamard product. Symmetry is $p(X,Y) \Rightarrow p(Y,X)$, so we assume both parts hold, i.e., $p(x,y) \wedge p(y,x)$, where $x$ and $y$ are not variables but specific entities. Applying the scoring function, we obtain $\mathbf{x} \circ \mathbf{p} = \mathbf{y} \wedge \mathbf{y} \circ \mathbf{p} = \mathbf{x}$, which holds if $\mathbf{p} \circ \mathbf{p} = \mathbf{1}$ [35].

In the table, we observe that there are many missing theoretical analyses, especially with regards to jointly captured patterns. Except Abboud et al. [1], the rest focus solely on patterns individually captured, i.e., they do not consider the presence of other patterns. Abboud et al. [1] studied patterns jointly captured. A type of pattern can aggregate multiple instances in a single graph, e.g., both $hyponym(X,Y) \Rightarrow hypernym(Y,X)$ and $meronym(X,Y) \Rightarrow holonym(Y,X)$ belong to the symmetry type. If a method individually captures all the instances of a pattern type without leading to contradiction, then the method jointly captures the pattern type.

Additionally, the analyses presented in the table do not study whether methods identify negatives properly. For instance, triples that fulfill $p(x,y) \wedge p(y',x)$ where $y \neq y'$ are negatives according to PCA [14]. In RotatE, $\mathbf{x} \circ \mathbf{p} = \mathbf{y} \wedge \mathbf{y'} \circ \mathbf{p} = \mathbf{x}$ such that $\mathbf{p} \circ \mathbf{p} = \mathbf{1}$, so $\mathbf{y} = \mathbf{y'}$ contradicts $\mathbf{y} \neq \mathbf{y'}$. RotatE cannot thus properly identify negatives for symmetry. Below, we discuss several embedding methods and briefly present their theoretical properties.

BoxE [1] embeds entities as real vectors and predicates as boxes. It captures various patterns, both independently and jointly, by deriving valid box configurations for each predicate. ComplEx [37] exploits complex numbers, and captures patterns under certain conditions, e.g., symmetry (antisymmetry) when the imaginary (real) parts of the predicate vectors are set to zero [1]. HAKE [41] combines a modulus and a phase parts. There is no theoretical analysis for HAKE. HolE [25] applies circular correlation to complex numbers. It is a bilinear method, so it individually captures hierarchy [1]. QuatE [40] uses quaternions with one real and three imaginary parts, and captures symmetry (antisymmetry) when the imaginary parts are set to (non-)zero. RotatE [35] embeds predicates as complex rotations between entities. It captures patterns under certain conditions like symmetry ($\mathbf{p} \circ \mathbf{p} = \mathbf{1}$). RotPro [33] extends RotatE by adding projections between entities. It captures the same patterns as RotatE and transitivity by applying consecutive orthogonal projections. TransE [7] uses a distance-based score in the Euclidean space. It also captures patterns under certain conditions like inversion ($\mathbf{p_1} = \mathbf{p_2}$). TorusE [13] generalizes TransE using different representations and operations. The authors focus on torus vectors. There is no theoretical analysis for TorusE.

Note that Table 1 only includes the embedding methods that are our focus in this paper. Other methods that study inference patterns are DensE [21], DualE [8], HousE [20], MDE [31], PairE [9], RatE [18] and Rotate3D [15], to mention a few examples.

## 2.2 Empirical studies

Rossi et al. [29] is the most related approach. The authors empirically studied the Hits@1 metric, how many triples in the test split are ranked first, for symmetry, antisymmetry and transitivity in WN18, WN18RR and YAGO3-10. The methods include ComplEx, HAKE, HolE, RotatE, TorusE and TransE. The main difference w.r.t. our approach is that we collectively quantify positive and negative evidence, while Hits@1 measures each triple in isolation. Also, we study a larger number of patterns with different characteristics, e.g., they contain single and multiple predicates.

Some studies aim to explain the inner workings of embedding methods. Abboud et al. [1] studied BoxE's box volumes in YAGO3-10, e.g., symmetric predicates have similar-sized boxes. Trouillon et al. [37] visualized ComplEx's vectors and found that predicates in WN18 were placed opposite to their inverse counterparts. Zhang et al. [41] and Sun et al. [35] studied histograms of HAKE's and RotatE's modulus and phase parts over few predicates in WN18, WN18RR and YAGO3-10. Nickel et al. [25] and Sun et al. [35] studied the capabilities of HolE and RotatE to capture transitivity and composition over a small dataset. Song et al. [33] studied RotPro's embedding phases in YAGO3-10 as well as its transitive capabilities w.r.t. RotatE for a single predicate. As a result, there is no consistent analysis on how patterns are captured by models. Also, quantifying how both positives and negatives are captured is missing.

Other empirical studies focus on different aspects tangentially related to inference patterns. Dettmers et al. [10] studied an "inverse model" over WN18 in which the inverse pattern was enforced. This "inverse model" achieves very good performance using rank-based metrics. Several authors have analyzed the rank-based results of 1-to-1, 1-to-many, many-to-1 and many-to-many predicates in WN18

**Figure 1: A sample knowledge graph partitioned into splits**

**Figure 2: Link prediction evaluation for** *lives*(*june*, *ny*) **in the test split in Figure 1b sorted by plausibility score, ascending**

or YAGO3-10 [2, 7, 13, 35, 40]. Studying predicate cardinality is different from analyzing inference capabilities.

## 3 Background

A knowledge graph $G$ is a set of $p(s, o)$ triples, where $s$ and $o$ are subject and object, respectively, and $p$ is predicate. $E$ denotes the set of entities in $G$, i.e., the union of subjects and objects. A model exploits numerical vectors in its scoring function $f$ to compute a plausibility score for $p(s, o)$, denoted as $f(p(s, o))$. We use $G_{TR}$, a subset of $G$, to learn the model, and $G_{VA}$, another subset of $G$, to evaluate early stopping such that $G_{TR} \cap G_{VA} = \emptyset$.

**Example 3.1.** Figure 1 presents a sample knowledge graph with triples modeling people working at companies (*works*) and living in cities (*lives*), as well as companies located in cities (*located*).

In link prediction evaluation, we use $G_{TE}$, a third subset of $G$ such that $G_{TR} \cap G_{TE} = \emptyset$ and $G_{VA} \cap G_{TE} = \emptyset$. Note that $G_{TR} \cup G_{VA} \cup G_{TE} = G$. Knowledge graphs typically operate under the open-world assumption, i.e., triples not present in $G$ are unknown [17]. However, learning a model requires negative examples. PCA [7, 11] deems negative triples those that are not present in $G$ when the subject or object, but not both, of a given triple is corrupted. The link prediction evaluation protocol takes each triple $p(s, o) \in G_{TE}$ as input and computes the following two ranks:

$$r_s(p(s, o), G) = 1 + |\{p(s', o) \mid s' \in E \land p(s', o) \notin G \land \\ f(p(s', o)) < f(p(s, o))\}| \quad (1)$$

$$r_o(p(s, o), G) = 1 + |\{p(s, o') \mid o' \in E \land p(s, o') \notin G \land \\ f(p(s, o')) < f(p(s, o))\}| \quad (2)$$

Intuitively, $r_s(p(s, o), G)$ is the rank (position) of $p(s, o)$ w.r.t. its subject corrupted counterparts when sorted by plausibility score.

Similarly, $r_o(p(s, o), G)$ compares object corrupted counterparts. Note that, for the sake of presentation, both $r_s(p(s, o), G)$ and $r_o(p(s, o), G)$ are optimistic ranks in which score ties are ignored; in practice, we use fractional ranks that help mitigate the presence of ties [5]. Accuracy is measured as aggregations of $r_s(p(s, o), G)$ and $r_o(p(s, o), G)$ like mean rank or mean reciprocal rank [36].

**Example 3.2.** In Figure 2, we corrupt the subject and object of $t_0 = lives(june, ny)$, and sort by the score assigned by a model in ascending order. In the example, $r_s(t_0, G) = 4$ and $r_o(t_0, G) = 2$. Note that $lives(mary, ny)$ is not a corrupted counterpart within $r_s(t_0, G)$ because it is present in Figure 1a.

A model should learn a number of inference patterns, i.e., link prediction rules that characterize the model's behavior. Each of these patterns is as follows: $l = \mathbb{B}(X, Y) \Rightarrow \mathbb{H}(X, Y)$, where $\mathbb{B}(X, Y)$ and $\mathbb{H}(X, Y)$ are the body and the head of the pattern, respectively. The body contains one or more atoms of the form $p(X, Y)$, where $X$ and $Y$ are variables, and $p$ is a predicate. If a body contains several atoms, they are composed using Boolean conjunction. The head of a pattern $\mathbb{H}(X, Y)$ can be either $p(X, Y)$, $p(Y, X)$ or $\neg p(Y, X)$.

**Example 3.3.** A composition pattern like $works(X, Z) \wedge located(Z, Y) \Rightarrow lives(X, Y)$ entails that, if $X$ works at $Z$ and $Z$ is located in $Y$, then $X$ lives in $Y$. An antisymmetry pattern like $works(X, Y) \Rightarrow \neg works(Y, X)$ means that, if $X$ works at $Y$, then there is no triple stating $Y$ works at $X$, which would be nonsensical.

## 4 Our method

Section 4.1 discusses core patterns in the original graph and how to quantify them. Section 4.2 describes our approach to collect predictions made by the model under evaluation. Section 4.3 presents our approach to assess patterns and measure how they are captured.

### 4.1 Core patterns

We assume there are several core patterns over $G$ of the form $l_i = \mathbb{B}(X, Y) \Rightarrow \mathbb{H}(X, Y)$. These core patterns can be mined from $G$ using rule mining methods [14, 22, 27]. These methods have different sets of supported rules, i.e., the regularities that they mine and ignore vary [22]. We can exploit multiple methods to find core patterns in practice as long as support and confidence can be measured. We use support to quantify positive evidence [14]:

$$supp(l_i, G) = \{(\phi(X), \phi(Y)) \mid \phi \in I(\mathbb{B}(X, Y) \wedge \mathbb{H}(X, Y), G)\} \quad (3)$$

where $I(\mathbb{A}, G)$ denotes the result of instantiating the set of atoms $\mathbb{A}$ over $G$. Each instantiation is an injective function $\phi : Var \rightarrow E$, where $Var$ is the set of variables in $\mathbb{A}$. Note that $Var$ always includes $X$ and $Y$, and can include extra variables as part of $\mathbb{B}(X, Y)$. Intuitively, $supp(l_i, G)$ is the set of pairs that are considered positive evidence by $l_i$, i.e., they fulfill the pattern.

**Example 4.1.** Let $l_0 = works(X, Z) \wedge located(Z, Y) \Rightarrow lives(X, Y)$ be a pattern, and $G = G_{TR} \cup G_{VA} \cup G_{TE}$ as in Figure 1. Support is $supp(l_0, G) = \{(bob, chi), (june, ny), (luca, ny)\}$. Note that $(mary, sf)$ and $(eden, sf)$ fulfill $l_0$'s body since both work at $wonka$ located in $sf$. However, they are not in the support because two triples state that $mary$ and $eden$ live in $ny$ and $chi$, respectively.

The pair of entities present in support are altered if we change the graph at hand. Let $G^{\star} = G_{TR} \cup G_{VA}$. Support is $supp(l_0, G^{\star}) = \{(bob, chi)\}$. Since $located(acme, ny)$ is in $G_{TE}$, $l_0$'s body and head are not fulfilled for $(june, ny)$ and $(luca, ny)$.

To quantify negative evidence, we can compute the pairs that fulfill only the body and remove positives from it, i.e., $\{(\phi(X), \phi(Y)) \mid \phi \in I(\mathbb{B}(X, Y), G)\} \setminus supp(l_i, G)$. This is how standard confidence is computed in rule mining [14]. The main issue is that it considers all non-existing pairs in $G$ as negatives, i.e., it operates under the closed-world assumption. We adopt the partial completeness assumption [7, 11, 14], and quantify negative evidence as follows:

$$neg(l_i, G) = \{(\phi(X), \phi(Y)) \mid \phi \in I(\mathbb{B}(X, Y) \wedge \mathbb{H}(X, Y'), G)\} \quad (4)$$

where $Y$ and $Y'$ cannot have the same instantiations, i.e., $Y \neq Y'$.

**Example 4.2.** The negative evidence of $l_0$ from Example 4.1 over $G$ in Figure 1 is as follows: $neg(l_0, G) = \{(mary, sf), (eden, sf)\}$. These are the pairs that satisfy the body of $l_0$ but not the head. Both $mary$ and $eden$ work at $wonka$ that is located in $sf$; however, they do not live in $sf$, but in $ny$ and $chi$, respectively. This entails that there are triples indicating where $mary$ and $eden$ live in $G$. As a result, these pairs are considered negative evidence.

### 4.2 Collecting predictions

Our goal is measure how a model has captured a given pattern. Therefore, we focus on the predictions made by the model at hand. However, to the best of our knowledge, there is no standard approach to collect these predictions. We propose to rely on link prediction evaluation. Specifically, using the evaluation protocol described above, we make the following observation:

**Observation 4.1.** In link prediction evaluation, triples ranked in the initial positions are the predictions that a model considers more plausible with high confidence. Each of these triples can be either a triple in the graph at hand or a corrupted counterpart.

We propose to collect triples ranked in the top $k$ during link prediction evaluation. However, we make the following observation:

**Observation 4.2.** In link prediction evaluation, a model considers not plausible the corrupted counterparts that are ranked below a triple in the knowledge graph at hand.

Therefore, even though a corrupted counterpart can be ranked in the top $k$, it is not collected if it is ranked below a triple in $G$.

**Example 4.3.** Assume we select $k = 5$ in Figure 2a. We collect the following four predictions: $lives(bob, ny)$, $lives(acme, ny)$, $lives(corp, ny)$ and $lives(june, ny)$. Since the latter triple is the one present in test, we stop collecting triples below it, i.e., $lives(luca, ny)$ is not collected even though its rank is 5. If we set $k = 2$, only two triples are collected: $lives(bob, ny)$ and $lives(acme, ny)$, and $lives(june, ny)$, the triple in $G$, is not collected.

Formally, we collect two sets of triples as follows:

$lives(bob, ny)$
$lives(acme, ny)$                                              $located(corp, ny)$
$lives(corp, ny)$            $lives(june, sf)$               $located(wonka, ny)$
$lives(june, ny)$            $lives(june, ny)$               $located(acme, ny)$

**(a)** $t_0$'s subject        **(b)** $t_0$'s object        **(c)** $t_1$'s subject

**Figure 3: Sets of triples collected for** $t_0 = lives(june, ny)$ **and** $t_1 = located(acme, ny)$ **in our running example**

$$R_s^k(p(s, o), G) = \{p(s', o) \mid s' \in E \wedge p(s', o) \notin G \wedge$$
$$r_s(p(s', o), G) \leq min(k, r_s(p(s, o), G))\} \cup \qquad (5)$$
$$\{p(s, o) \text{ if } r_s(p(s, o), G) \leq k\}$$

$$R_o^k(p(s, o), G) = \{p(s, o') \mid o' \in E \wedge p(s, o') \notin G \wedge$$
$$r_o(p(s, o'), G) \leq min(k, r_o(p(s, o), G))\} \cup \qquad (6)$$
$$\{p(s, o) \text{ if } r_o(p(s, o), G) \leq k\}$$

Each corrupted counterpart $p(s', o)$ in $R_s^k(p(s, o), G)$ fulfills that its rank $r_s(p(s', o), G)$ is less or equal than the minimum between $k$ and the rank of $p(s, o)$. $R_s^k(p(s, o), G)$ contains $p(s, o)$ if its rank is less or equal than $k$. The same applies to $R_o^k(p(s, o), G)$. The collected triples for link prediction evaluation are as follows:

$$G_{PR}^k = \bigcup_{p(s,o) \in G_{TE}} R_s^k(p(s, o), G) \cup R_o^k(p(s, o), G) \qquad (7)$$

**Example 4.4.** Assume we set $k = 5$ in Figures 2a and 2b. For $t_0 = lives(june, ny)$, the sets of collected triples $R_s^{k=5}(t_0, G)$ and $R_o^{k=5}(t_0, G)$ are presented in Figures 3a and 3b, respectively. In this example, $t_0$ is collected in both sets since its rank is less than 5 in both cases. For $t_1 = located(acme, ny)$, assume we have the $R_s^{k=5}(t_1, G)$ set depicted in Figure 3c, and $R_o^{k=5}(t_1, G) = \{t_1\}$. As a result, $G_{PR}^{k=5}$, the set of collected predictions, is the union of the sets of triples in Figures 3a, 3b and 3c.

### 4.3 Pattern assessment

Given a core pattern $l_i = \mathbb{B}(X, Y) \Rightarrow \mathbb{H}(X, Y)$, we aim to quantify how it is captured by a model. We compare positive and negative evidence obtained for $l_i$ over $G$ w.r.t. evidence obtained over $G'^k = G_{TR} \cup G_{VA} \cup G_{PR}^k$ using a set similarity measure as follows:

$$\pi^k(l_i) = sim(supp(l_i, G), supp(l_i, G'^k)) \qquad (8)$$
$$v^k(l_i) = sim(neg(l_i, G), neg(l_i, G'^k)) \qquad (9)$$

where $\pi^k(l_i)$ and $v^k(l_i)$ are positive and negative evidence, respectively, and $sim(A, B)$ is a similarity measure between sets $A$ and $B$ like Jaccard, $\mathcal{J}(A, B) = |A \cap B|/|A \cup B|$, or Sørensen–Dice, $DS(A, B) = 2|A \cap B|/(|A| + |B|)$.

**Example 4.5.** For pattern $l_0 = works(X, Z) \wedge located(Z, Y) \Rightarrow lives(X, Y)$ and $G'^{k=5} = G_{TR} \cup G_{VA} \cup G_{PR}^{k=5}$, where $G_{TR}$ and $G_{VA}$ are respectively presented in Figures 1a and 1b, and $G_{PR}^{k=5}$ in Example 4.4, we get the following sets: $supp(l_0, G'^{k=5}) = \{(bob, chi), (june, ny),$

$(luca, ny), (bob, ny), (mary, ny)\}$ and $neg(l_0, G'^{k=5}) = \{(mary, sf), (eden, sf), (eden, ny)\}$. Assuming Jaccard similarity, we obtain the following measurements for positive and negative evidence, respectively: $\pi^{k=5}(l_0) = 3/5$ and $v^{k=5}(l_0) = 2/3$. The sets computed for $G$, $supp(l_0, G)$ and $neg(l_0, G)$, are respectively presented in Examples 4.1 and 4.2. Note that the sets have many triples in common.

As illustrated in the previous example, there are many common triples between $supp(l_i, G)$ and $supp(l_i, G'^k)$, and between $neg(l_i, G)$ and $neg(l_i, G'^k)$. This is because, typically, $|G^\star| = |G_{TR} \cup G_{VA}| \gg |G_{TE}|$. We wish to reduce the effect of $G^\star$, so we propose to correct the similarity between sets as follows:

$$\pi_c^k(l_i) = sim_c(supp(l_i, G), supp(l_i, G'^k), supp(l_i, G^\star)) \qquad (10)$$
$$v_c^k(l_i) = sim_c(neg(l_i, G), neg(l_i, G'^k), neg(l_i, G^\star)) \qquad (11)$$

where $sim_c(A, B, C)$ is a corrected set similarity as follows:

$$sim_c(A, B, C) = sim(A \setminus C, B \setminus C) \qquad (12)$$

**Example 4.6.** For pattern $l_0 = works(X, Z) \wedge located(Z, Y) \Rightarrow lives(X, Y)$ and $G^\star = G_{TR} \cup G_{VA}$, where $G_{TR}$ and $G_{VA}$ are respectively presented in Figures 1a and 1b, we get the following sets: $supp(l_0, G^\star) = \{(bob, chi)\}$ and $neg(l_0, G^\star) = \{(mary, sf), (eden, sf)\}$. As a result, following the previous examples, we have that $\pi_c^{k=5}(l_0) = 2/3$ and $v_c^{k=5}(l_0) = 0/1$ assuming Jaccard similarity. Note that these values were $\pi^{k=5}(l_0) = 3/5$ and $v^{k=5}(l_0) = 2/3$ without correction, which illustrates its effect.

## 5 Experiments

Section 5.1 presents datasets and models we trained. Section 5.2 discusses our experimental setup. Sections 5.3 and 5.4 respectively analyze existing and new inference patterns that we introduce.

### 5.1 Datasets and models

We used the several datasets that are common in link prediction evaluation [2, 5, 6, 29, 30, 32, 36]. These datasets are already partitioned into training, validation and test splits:

|          | $|E|$    | $|P|$ | $|G_{TR}|$  | $|G_{VA}|$ | $|G_{TE}|$ |
| -------- | -------- | ----- | ----------- | ---------- | ---------- |
| WN18     | 40,943   | 18    | 141,442     | 5,000      | 5,000      |
| WN18RR   | 40,943   | 11    | 86,835      | 3,034      | 3,134      |
| YAGO3-10 | 123,182  | 37    | 1,079,040   | 5,000      | 5,000      |

WN18 deals with English words and was extracted from Word-Net [23]. WN18RR is similar to WN18 but inverse predicates have been removed [10], e.g., WN18RR contains *meronym* but not *holonym*. Capturing a pattern like $meronym(X, Y) \Rightarrow holonym(Y, X)$ is challenging. YAGO3-10 [34] contains triples related to people and places, and was extracted from Wikipedia and unified using WordNet.

We trained one model for each dataset[1] using the methods presented in Section 2, namely: BoxE [1], ComplEx [37], HAKE [41], HolE [25], QuatE [40], RotatE [35], RotPro [33], TorusE [13] and TransE [7]. They exploit a variety of numerical vectors and scoring functions. We used a combination of a Sobol sequence and a

---

[1]Models, source code and results are publicly available: double-blind review.

**Table 2: Arithmetic mean of the positive evidence ($\mu_\pi$) using $k = 5$ with best results in bold. In each cell, up and down arrows entail whether or not a pattern is independently or jointly captured. Colored cells indicate unexpected results.**

| | | BoxE | | ComplEx | | HAKE | | HolE | | QuatE | | RotatE | | RotPro | | TorusE | | TransE | |
|---|---|---|---|---|---|---|---|---|---|---|---|---|---|---|---|---|---|---|---|
| WN18 | Symmetry | .91 | ⇑ | .87 | ⇑ | .77 | – | .78 | – | **.94** | ↑ | .80 | ⇑ | .68 | ↑ | .58 | – | .58 | ⇓ |
| | Antisymmetry | .44 | ⇑ | .29 | ⇑ | .28 | – | .26 | – | .53 | ↑ | .37 | ⇑ | **.56** | ↑ | .44 | – | .45 | ⇑ |
| | Inversion | .91 | ⇑ | .85 | ⇑ | .86 | – | .83 | – | .92 | ↑ | .89 | ⇑ | **.93** | ↑ | .90 | – | .87 | ↕ |
| | Transitive | .56 | – | **.64** | ↓ | .19 | – | .15 | – | .54 | ↓ | .15 | ↓ | .11 | ↑ | .09 | – | .09 | ↓ |
| WN18RR | Symmetry | .61 | ⇑ | .92 | ⇑ | .34 | – | .82 | – | **.94** | ↑ | .90 | ⇑ | .66 | ↑ | .58 | – | .58 | ⇓ |
| | Antisymmetry | .09 | ⇑ | .03 | ⇑ | .00 | – | .02 | – | **.12** | ↑ | .11 | ⇑ | .03 | ↑ | .05 | – | .05 | ⇑ |
| | Transitive | .15 | – | .60 | ↓ | .24 | – | .34 | – | **.81** | ↓ | .38 | ↓ | .11 | ↑ | .09 | – | .09 | ↓ |
| YAGO3-10 | Hierarchy | .14 | ⇑ | **.25** | ↕ | .16 | – | .14 | ↕ | .21 | – | .16 | ⇓ | .15 | – | .20 | – | .10 | ⇓ |
| | Symmetry | .18 | ⇑ | .17 | ⇑ | **.40** | – | .11 | – | .30 | ↑ | .24 | ⇑ | .20 | ↑ | .28 | – | .10 | ⇓ |
| | Antisymmetry | .07 | ⇑ | .05 | ⇑ | .05 | – | .01 | – | .06 | ↑ | .02 | ⇑ | .02 | ↑ | .03 | – | **.08** | ⇑ |
| | Inversion | .12 | ⇑ | **.49** | ⇑ | .31 | – | .22 | – | .09 | ↑ | .35 | ⇑ | .33 | ↑ | .21 | – | .09 | ↕ |
| | Transitive | .06 | – | .08 | ↓ | .09 | – | .07 | – | .02 | ↓ | **.12** | ↓ | .08 | ↑ | .08 | – | .05 | ↓ |
| | Composition | .24 | ⇓ | .14 | ⇓ | .25 | – | .15 | – | .23 | – | .18 | ↕ | .15 | ↑ | .23 | – | **.26** | ↕ |

Bayesian optimizer to find the best configuration of hyperparameter values [36]. For a fair comparison, we fixed the size of the embedding vectors to 150. All methods were implemented within the same framework, and all models were trained using the same triple batches during stochastic gradient descent. We trained the models for 1,000 epochs, and used the mean rank over the validation split as early stopping criteria every 50 epochs. We used PCA to generate corrupted counterparts during training and validation.

## 5.2 Experimental setup

We used AMIE [14] to mine core patterns from each dataset. We used a minimum head coverage of .10 and PCA confidence of .10. Head coverage is the size of the positive evidence divided by the total number of triples with the head predicate. PCA confidence is the size of the positive evidence divided by the sizes of positive plus negative evidence. These core patterns were of the following types: hierarchy, symmetry, inversion, transitive and composition. Antisymmetry core patterns were mined manually for each predicate present in the dataset. Intersection core patterns were also mined manually: we combined every two hierarchy patterns with the same head predicate. Mined core patterns are as follows:

| | WN18 | WN18RR | YAGO3-10 |
|---|---|---|---|
| Hierarchy | 0 | 0 | 11 |
| Symmetry | 3 | 3 | 4 |
| Antisymmetry | 18 | 11 | 34 |
| Inversion | 14 | 0 | 4 |
| Intersection | 0 | 0 | 0 |
| Transitive | 1 | 1 | 3 |
| Composition | 0 | 0 | 12 |
| Gen. Intersection | 0 | 0 | 3 |
| B. Transitive | 1 | 1 | 1 |
| Equality | 1 | 1 | 4 |
| B. Composition | 2 | 0 | 0 |
| Commonality | 2 | 0 | 4 |

Note that AMIE mined additional patterns like generic intersection or backward transitive that are introduced below.

To present our results, we group patterns by type and compute the arithmetic mean of the positive evidence using $k = 5$, i.e., $\pi_c^{k=5}(l_i)$. We denote this as $\mu_\pi$. This value of $k$ was the one we empirically determined to be the best for all the datasets under evaluation. Similarly, we report the arithmetic mean of the negative evidence, $v_c^{k=5}(l_i)$, and denote it as $\mu_v$. We use Sørensen–Dice to measure set similarity, since we believe it fits better in our context: a model should be allowed to have a certain deviation from the expected results; otherwise, the model is overfitted. We discuss below how results change when we set $k = 10$.

## 5.3 Existing patterns

Table 2 presents our quantitative results for positive evidence over the inference patterns studied above. We summarize the theoretical results in the table as follows: a double up (down) arrow means a pattern is (not) expected to be captured both independently and jointly; a single up (down) arrow entails a pattern is (not) expected to be captured independently and jointly is unknown; an up–down arrow means yes independently but not jointly; dash means unknown. In WN18, the ComplEx and RotatE models exhibit lower values than expected for the antisymmetry pattern. For the inversion pattern, TransE achieves a high value even though it is not expected to jointly capture the pattern. Similarly, ComplEx and QuatE are not expected to independently capture the transitive pattern, but they exhibit high values. However, RotPro, which captures transitivity, exhibits a low value. We hypothesize these unexpected results are due to the high presence of redundancy in WN18.

In WN18RR with reduced redundancy, we generally observe lower values than in WN18, which is expected. This reduction is quite significant for the antisymmetry pattern, which all models struggle to capture. In the other patterns, BoxE and HAKE are the ones that have a significant performance drop w.r.t. WN18. QuatE is the best performing model in both WN18 and WN18RR. Also, TorusE performs very similar to TransE. Several models achieve better values than expected for the transitive pattern. Comparing the values achieved for WN18 vs. WN18RR, we observe the performance of many models remain or increase, even significantly

like QuatE, but BoxE's performance drops. Our results suggest that inverse predicates significantly help increase performance for the antisymmetry and transitive patterns.

YAGO3-10 is a challenging dataset for link prediction, so overall values are lower than those achieved in the WordNet datasets. We observe that BoxE is not among the best models except for the composition pattern, which is not expected to capture. This competitive performance for the composition pattern was also reported by BoxE's authors [1]. Overall, ComplEx and HAKE are the best models. It is surprising that ComplEx and TransE respectively achieve the best results for the hierarchy and composition patterns, even though they are not able to jointly capture them.

We compare our results with those reported by Rossi et al. [29] that studied symmetry, antisymmetry and transitivity. Note that they trained a completely different set of models than us. They used Hits@1, i.e., how many test triples are ranked first, which is more restrictive than our setting of $k = 5$. We observe very similar results, even the same general performance drop for the antisymmetry pattern in WN18RR. Also, their TransE models achieve .00 for the symmetry pattern in the three datasets. We achieve the same results when $k = 1$. However, we observe a few differences as follows:

- Except for antisymmetry in WN18RR, all their models achieve higher values compared to ours. Our hypothesis is that they set a .50 tolerance threshold, similar to setting the minimum head coverage to .50. Our tolerance threshold is .10.
- Their results for antisymmetry in both WN18 and YAGO3-10 are much higher than our results, in which all of our models struggle. This can be explained again because of the different tolerance threshold used.
- For the transitive pattern in YAGO3-10, all of their models achieve similar results except TransE, which is the worst performing. We can observe the same performance in our results, but, as mentioned earlier, with lower values.

Furthermore, we study the effect of increasing from $k = 5$ to $k = 10$.[2] In general, all values remain or drop for all patterns and models when we increase $k$. The models more benefited are, in WN18, BoxE, ComplEx and QuatE with an increase of .19 or more for the transitive pattern; HolE and RotatE in WN18RR with an increase of .11 or more for the transitive pattern; and RotPro in YAGO3-10 for the inversion pattern with an increase of .14.

Besides positive, we also study negative evidence.[3] All models achieve very low values for all patterns, except for the transitive pattern in WN18 and WN18RR. RotPro, QuatE and RotatE respectively achieve approximately .60 and .49 in both cases, while HAKE is the worst performing model with .06 or less. However, these values are generally lower than those achieved for the positive evidence. This suggests that the models generally struggle to identify negative evidence, which entails that they do not adhere to PCA, even though they were trained using it. Finally, we studied the effect of increasing from $k = 5$ to $k = 10$; all negative evidence values using $k = 10$ were lower than those achieved using $k = 5$.

***Takeaways.*** In the WordNet datasets, redundancy makes models improve performance, even though they are not expected to

capture a pattern. Increasing inverse triples help several models better capture antisymmetry and transitivity. QuatE performs best overall. In the challenging datasets (WN18RR and YAGO3-10), even though BoxE is expected to jointly capture many patterns, it is not among the best performing models. The exception is the composition pattern, which BoxE cannot capture. In general, increasing from $k = 5$ to $k = 10$ decreases performance. Regarding negative evidence, all models generally exhibit low or very low values with very few exceptions. This suggests that these models fail to follow the partial completeness assumption.

## 5.4 New patterns

There are core patterns of interest that do not adhere to the pattern definitions described above. We focus on the following:

**Generic intersection.** Like intersection, it only has two variables in total. However, we allow the variables in the body to be connected in any arbitrary way. For example, in a body like $p_1(X, Y) \wedge p_2(Y, X)$, the intersection occurs between the objects of $p_1$ and the subjects of $p_2$. We also do not enforce that all predicates must be different. As an example, in YAGO3-10, $deals(X, Y) \wedge deals(Y, X) \Rightarrow neighbor(X, Y)$, where $deals(x, y)$ means $x$ deals with $y$ in a broader sense.

**Backward transitive.** It is a transitive pattern in which the body forms a backward path, i.e., $p(Y, Z) \wedge p(Z, X)$. For instance, in WN18 and WN18RR, $seeAlso(Y, Z) \wedge seeAlso(Z, X) \Rightarrow seeAlso(X, Y)$.

**Equality.** In the transitive and backward transitive patterns, we assume the body forms a forward or a backward path, respectively. In the equality pattern, the body does not form a path, i.e., $p(X, Z) \wedge p(Y, Z)$, where $Z$ is sink, or $p(Z, X) \wedge p(Z, Y)$, where $Z$ is source. Note that, when transitive, backward transitive and equality occur at the same time between the same entities, it means that the entities are equal, which is the reason for this pattern's name. In practice, each of these transitive, backward transitive and equality patterns achieves different results for positive and negative evidence due to missing triples. For example, in YAGO3-10, $neighbor(X, Z) \wedge neighbor(Y, Z) \Rightarrow neighbor(X, Y)$.

**Backward composition.** It is a composition pattern such that the body forms a backward path, i.e., $p_1(Y, Z) \wedge p_2(Z, X)$. For example, in WN18, $topic(Y, Z) \wedge hyponym(Z, X) \Rightarrow synonym(X, Y)$, where $topic(y, z)$ means $z$ is a scientific category of $y$.

**Commonality.** It is similar to the equality pattern but applied to composition. That is, it is a composition in which the body is neither a forward nor a backward paths. Therefore, variable $Z$ in the body is a sink or a source like in the equality pattern. The name is because $Z$ is an entity common to two other entities. For example, in YAGO3-10, $died(X, Z) \wedge capital(Y, Z) \Rightarrow citizen(X, Y)$.

Table 3 presents positive and negative evidence results for these new patterns. Regarding positive evidence, for presentation purposes, we refer to the closest known theoretical analysis for each new pattern. We use the intersection pattern to analyze generic intersection, the transitive pattern to analyze backward transitive and equality, and the composition pattern to analyze backward composition and equality. In WN18, ComplEx's performance is surprising, since it is unable to capture neither the transitive nor the composition patterns. Similarly, QuatE cannot capture the transitive pattern. These results coincide with our results for the transitive pattern over WN18. The BoxE and RotatE models also achieve surprising

---

[2]These results are reported elsewhere: double-blind review
[3]These results are reported elsewhere: double-blind review

**Table 3: Arithmetic mean of the positive evidence ($\mu_\pi$) using $k = 5$ with best results in bold, and negative evidence ($\mu_v$) in parentheses for the new inference patterns. Colored cells indicate surprising results w.r.t. the known theoretical analyses.**

|  |  | BoxE | ComplEx | HAKE | HolE | QuatE | RotatE | RotPro | TorusE | TransE |
|---|---|---|---|---|---|---|---|---|---|---|
| WN18 | B. Transitive | **.71** (.45) | .61 (.40) | .27 (.13) | .22 (.31) | .69 (.48) | .28 (.35) | .15 (.42) | .13 (.26) | .14 (.30) |
|  | Equality | **.57** (.31) | .53 (.29) | .13 (.17) | .17 (.26) | **.57** (.44) | .18 (.40) | .12 (.45) | .10 (.29) | .11 (.34) |
|  | B. Composition | .84 (.37) | .77 (.05) | .92 (.11) | .84 (.09) | **.93** (.32) | .90 (.28) | .79 (.50) | .41 (.28) | .38 (.20) |
|  | Commonality | .84 (.36) | .77 (.05) | .91 (.10) | .85 (.09) | **.94** (.31) | .90 (.28) | .79 (.49) | .42 (.27) | .37 (.20) |
| WN18RR | B. Transitive | .17 (.08) | .66 (.38) | .07 (.06) | .42 (.30) | **.76** (.50) | .51 (.43) | .17 (.39) | .14 (.27) | .13 (.24) |
|  | Equality | .17 (.07) | .59 (.32) | .26 (.10) | .32 (.27) | **.84** (.38) | .43 (.28) | .13 (.45) | .11 (.33) | .10 (.28) |
| YAGO3-10 | Gen. Intersection | .37 (.01) | **.76** (.01) | .28 (.00) | .47 (.17) | .22 (.00) | .48 (.00) | .51 (.00) | .36 (.00) | .23 (.33) |
|  | B. Transitive | .15 (.05) | .30 (.10) | .26 (.15) | .20 (.04) | .00 (.00) | **.36** (.31) | .29 (.20) | .27 (.14) | .21 (.00) |
|  | Equality | .12 (.03) | .15 (.05) | .14 (.08) | .10 (.03) | .09 (.01) | **.18** (.17) | .13 (.09) | .13 (.06) | .11 (.01) |
|  | Commonality | .09 (.04) | .06 (.03) | .00 (.03) | .01 (.04) | .07 (.03) | .07 (.04) | .00 (.04) | .06 (.04) | **.15** (.19) |

results for backward composition and commonality. RotPro is able to capture transitivity, but it exhibits low values for both backward transitivity and equality. The RotPro model also achieves low values for the transitive pattern. Note that, in this case, the backward composition and the equality patterns are the same but with different connections in the body (backward path vs. sources or sinks). We observe the same results in WN18RR with the main exception that RotatE achieves competitive results. Note that RotatE was not among the best performing models for the transitive pattern.

In YAGO3-10, the BoxE model struggles to achieve competitive results for the generic intersection pattern, even though it is able to capture the intersection pattern. Also, the BoxE model achieves the second-best result for the composition pattern, but it is not among the best performing models for the commonality pattern, which is expected since BoxE is unable to capture composition. Other surprising results are the values obtained by the ComplEx and RotatE models, which cannot capture neither intersection nor transitivity. The TransE model exhibits the best performance for commonality, even though it cannot capture composition. We observe the same results for the composition pattern.

Regarding negative evidence, we generally observe much higher values than those achieved for the existing patterns. We hypothesize that this is because several of these new patterns are related to transitivity, which was the pattern achieving the best results for negative evidence in the existing patterns. Still, many models struggle to identify negative evidence in the backward composition and commonality patterns.

***Takeaways.*** Several models achieve better positive evidence values than expected, even improving the results obtained for the existing patterns. Using the closest known theoretical analysis, many of the observed results for the existing patterns are also fulfilled for the new patterns. Even though several models are able to better capture negative evidence in these new patterns, many models still struggle to learn the partial completeness assumption.

## 6 Conclusions

We present a method-agnostic approach to empirically assess how models capture inference patterns in knowledge graphs. For each pattern, we compare both positive and negative evidence between the ground truth, the whole graph at hand, and the top-$k$ predictions

made by a model as part of link prediction evaluation. Based on our results, we make several remarks.

First, our experimental results only apply to the models we trained. Even though many of our observations agree with other observations found in the literature, they are empirical and, therefore, do not generally apply to the embedding methods. For example, a BoxE or a TorusE models trained differently may achieve different results. As future work, we aim to study the impact of different training options on a model's ability to capture inference patterns. These include hyperparameter values [3], assumptions to identify corrupted counterparts [4], and injecting inference patterns [1].

Second, we have detected the inability of several models to accurately capture negative evidence, which, as far as we know, has been studied for the first time. Despite applying the partial completeness assumption to train the models, they fail to learn the assumption, which explains the poor during link prediction evaluation. We hypothesize this is because, during training, corrupted counterparts were randomly selected. As future work, we will explore strategies to generate corrupted counterparts guided by patterns, which can be mined from the training split as a pre-processing step.

Third, in YAGO3-10, we observe that, for a given inference pattern, a substantial portion of both positive and negative evidence is present in the training and validation splits. This is not as exacerbated in WN18RR. This observation implies that predicting evidence in YAGO3-10 is extremely challenging. We hypothesize this issue can be mitigated by carefully selecting the graph splits. As a future work, we plan to explore graph partitioning algorithms guided by patterns. These algorithms will consider not only graph-based properties like indegrees and outdegrees [36], but also that evidence derived from the test split is relevant.

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

# A  Qualitative analysis

Our goal in this section is to shed light on some of the quantitative results obtained for WN18, WN18RR and YAGO3-10. First, we focus on the comparison between WN18 and WN18RR. A symmetry pattern that is common to both datasets is as follows:

$$verbGroup(Y, X) \Rightarrow verbGroup(X, Y)$$

where *verbGroup* establishes that two verbs have similar meaning. The pattern thus implies that the two words are similar to each other. Comparing $\pi_c^{k=5}$ values for WN18 and WN18RR, we observe that QuatE performs the best (0.99) on both datasets, indicating that the models are able to infer this pattern with the same level of accuracy, even after removal of inverse triples. This is expected as QuatE captures the symmetry pattern. TorusE achieves the lowest value in WN18 (.60), and HAKE in WN18RR (.60). The models most impacted by the removal of triples are BoxE (from .97 to .62) and HAKE (from .85 to .21). Interestingly, both ComplEx and RotatE achieve higher values in WN18RR than in WN18. We observe the same behavior in Table 2. RotPro, TorusE and TransE are the worst performing models achieving values less or equal than 0.7.

Another symmetry pattern is as follows:

$$alsoSee(Y, X) \Rightarrow alsoSee(X, Y)$$

Comparing $\pi_c^{k=5}$ values for WN18 and WN18RR, QuatE achieves the best values (.90 and .88, resp.), and HAKE achieves the lowest values (.50 and .10, resp.) among all models. BoxE (.83 and .64, resp.) and HAKE are the models most impacted by the redundancy removal. RotatE (.69 and .82, resp.) exhibits the highest increase.

In YAGO3-10, we first focus on the hierarchy pattern. A sample pattern is as follows:

$$isAffiliatedTo(X, Y) \Rightarrow playsFor(X, Y)$$

The pattern states that, if $X$ is affiliated to team $Y$, then $X$ plays for team $Y$. HAKE achieves the highest $\pi_c^{k=5}$ value (.51), whereas TransE achieves the lowest (.10). Although BoxE is theoretically able to capture the hierarchy pattern, both independently and jointly, the model achieves the second lowest values among all models (.21).

A sample transitive pattern is as follows:

$$isLocatedIn(X, Z) \wedge isLocatedIn(Z, Y) \Rightarrow islocatedIn(X, Y)$$

The *isLocatedIn* predicate between two entities $X$ and $Y$ states that $X$ is located in $Y$. Although this pattern best represents the transitive property, the $\pi_c^{k=5}$ values are generally low, less or equal than .10, indicating that all models struggle to accurately capture the pattern. RotPro, which captures transitivity, achieves the lowest value among all models (.02).

A sample symmetry pattern is as follows:

$$isConnectedTo(Y, X) \Rightarrow isConnectedTo(X, Y)$$

HAKE achieves the best $\pi_c^{k=5}$ value among all models (.27). TransE achieves the lowest value among all models (.11), which is expected as TransE is unable to capture symmetry. Although several other models are able to capture the symmetry pattern, we observe that the values are less or equal than .30 for all models

under evaluation, indicating that the models struggle to capture the pattern accurately.

# B  Extra datasets

BioKG [38] and Hetionet [16] contain biomedical knowledge, such as proteins, genes, diseases, drugs and their interactions. The datasets are as follows:

|          | $|E|$   | $|P|$ | $|G_{TR}|$ | $|G_{VA}|$ | $|G_{TE}|$ |
|----------|---------|-------|-----------|-----------|-----------|
| BioKG    | 105,524 | 17    | 2,057,658 | 5,170     | 5,170     |
| Hetionet | 45,158  | 24    | 2,238,946 | 5,626     | 5,625     |

The core patterns mined are as follows:

|                    | BioKG | Hetionet |
|--------------------|-------|----------|
| Hierarchy          | 14    | 8        |
| Symmetry           | 2     | 2        |
| Antisymmetry       | 34    | 34       |
| Inversion          | 0     | 0        |
| Intersection       | 6     | 2        |
| Transitive         | 2     | 2        |
| Composition        | 2     | 26       |
| Gen. Intersection  | 0     | 0        |
| B. Transitive      | 0     | 0        |
| Equality           | 4     | 4        |
| B. Composition     | 0     | 0        |
| Commonality        | 12    | 30       |

Tables 4 and 5 present our quantitative results for the datasets.

**Table 4: Arithmetic mean of the positive evidence ($\mu_\pi$) using $k$ = 5 with best results in bold**

|  |  | BoxE | ComplEx | HAKE | HolE | QuatE | RotatE | RotPro | TorusE | TransE |
|---|---|---|---|---|---|---|---|---|---|---|
| BioKG | Hierarchy | .19 ⇑ | .40 ↕ | **.46** – | .30 ↕ | .39 – | .45 ⇓ | .27 – | .33 – | .41 ⇓ |
|  | Symmetry | .00 ⇑ | .05 ⇑ | .03 – | **.07** – | .04 ↑ | **.07** ⇑ | **.07** ↑ | .06 – | .06 ⇓ |
|  | Antisymmetry | .04 ⇑ | .13 ⇑ | .21 – | .09 – | .15 ↑ | **.17** ⇑ | .10 ↑ | .09 – | .15 ⇑ |
|  | Intersection | .19 ⇑ | **.51** ⇓ | .47 – | .37 – | .42 – | .49 ↕ | .36 – | .32 – | .44 ↕ |
|  | Transitive | .03 – | .18 ↓ | .19 – | .21 – | .15 ↓ | .23 ↓ | **.27** ↑ | .11 – | .05 ↓ |
|  | Composition | .44 ⇓ | .47 ⇓ | .53 – | .56 – | .66 – | .63 ↕ | **.70** ↑ | .39 – | .43 ↕ |
| Hetionet | Hierarchy | .20 ⇑ | .07 ↕ | .20 – | .12 ↕ | .09 – | **.21** ⇓ | **.21** – | .15 – | .17 ⇓ |
|  | Antisymmetry | .05 ⇑ | .05 ⇑ | .07 – | .02 – | .05 ↑ | **.08** ⇑ | .07 ↑ | .04 – | .05 ⇑ |
|  | Intersection | .24 ⇑ | .08 ⇓ | .23 – | .13 – | .11 – | .26 ↕ | **.27** – | .17 – | .21 ↕ |
|  | Transitive | **.09** – | .08 ↓ | **.09** – | .06 – | .07 ↓ | .07 ↓ | .05 ↑ | .05 – | .05 ↓ |
|  | Composition | **.18** ⇓ | .07 ⇓ | .12 – | .09 – | .07 – | .11 ↕ | .15 ↑ | .09 – | .14 ↕ |

**Table 5: Arithmetic mean of the positive evidence ($\mu_\pi$) using $k$ = 5 with best results in bold and negative evidence ($\mu_\nu$)**

|  |  | BoxE | ComplEx | HAKE | HolE | QuatE | RotatE | RotPro | TorusE | TransE |
|---|---|---|---|---|---|---|---|---|---|---|
| BioKG | Equality | .03 (.04) | .18 (.03) | .21 (.13) | .20 (.09) | .14 (.01) | .20 (.10) | **.25** (.08) | .10 (.04) | .05 (.02) |
|  | Commonality | .25 (.06) | .43 (.15) | .40 (.14) | .37 (.12) | .41 (.11) | **.44** (.19) | .39 (.14) | .34 (.15) | .38 (.16) |
| Hetionet | Equality | **.09** (.04) | .08 (.04) | .07 (.04) | .07 (.02) | .06 (.03) | .07 (.05) | .05 (.03) | .03 (.02) | .06 (.02) |
|  | Commonality | **.17** (.11) | .05 (.02) | .14 (.12) | .07 (.02) | .05 (.02) | .12 (.07) | .14 (.09) | .08 (.04) | .12 (.08) |