# OpenReview forum: "A Method for Assessing Inference Patterns Captured by Embedding Models in Knowledge Graphs"
_ACM.org/TheWebConf/2024/Conference — TheWebConf24 Oral_

### Official Review · Reviewer_KMi3 · 2023-11-17

**Novelty:** 7
**Technical Quality:** 6

**Review:**

In this study, the authors experimentally investigate the inference patterns that can be learned by Knowledge Graph embedding models. 9 different Knowledge Graph embedding models are discussed. The properties of each model are clarified based on the experimental results. This research clarifies the current state and challenges in Knowledge Graph embedding research, and is expected to have a significant impact on the development of future research.

Pros:
Identifies properties of various Knowledge Graph embedding models
Identifies inference patterns that are difficult to learn

Cons:
Selection criteria of the used inference patterns could be change
Theoretical analysis is also desirable

I highly evaluate this study as an important analysis for the development of future research on knowledge graph embedding. Although actual data was used in the experiments, I think that the properties of the models would become clearer if synthetic data were also used in the analysis of inference patterns that can be learned. We look forward to further reports in the future.

In this study, we used patterns extracted from AMIE. It would be better to have a discussion on the validity of the inference patterns produced by AMIE, or to use other pattern extraction methods as well.

**Questions:**

What criteria were used to select the New Pattern in section 5.4? Does this pattern cover all the patterns extracted by AIME?

Thank you very much for your comments.

**Reviewer Confidence:**

3: The reviewer is confident but not certain that the evaluation is correct

**Scope:**

4: The work is relevant to the Web and to the track, and is of broad interest to the community

---

### Official Review · Reviewer_kihP · 2023-11-21

**Novelty:** 6
**Technical Quality:** 6

**Review:**

This paper tries to empirically determine the strengths and weaknesses of various knowledge graph embedding methods for capturing logical inference patterns in a knowledge graph (e.g., inversion, transitivity, symmetry etc in relations).  The core idea is to group different relations by type, and combine it with existing theoretical analyses that show if the embedding method jointly or independently capture the patterns to make inferences about weaknesses in methods.  Also important is the introduction of negative triples using the partial completeness assumption, and determining if the model has learnt to identify negative triples according to PCA.  I think this is a nice piece of empirical work, but I do have questions about it, and think it can be strengthened significantly.

**Questions:**

1.  Perhaps my most serious concern is the number of triples that constituted each pattern over which the results were computed. I am referring here to the unnamed table which defines the mined core patterns.  My understanding is that this is the base on which the results were computed.  For many patterns, N is under 10, and even in the single best case of antisymmetry the total number is under 100.  This makes me wonder how valid the results are, how generalizable, etc.  Is there any way to increase this?  Is it possible to synthetically create a knowledge graph that has the patterns and then test what happens to each embedding method?  I realize this is a big ask, but right now, these numbers make it hard to interpret anything.
2.  Related to 1, above, could you describe how you split these specific examples into train/test/valid - this is especially serious with the small numbers in each category.
3.  The writing is generally quite good, but I thought that the whole discussion of jointly/independently embedded needed a bit more explanation to make the paper more self contained, as was the discussion of the different arrows in the results.  Also confusing was the discussion of how things varied from 'expected'.  For instance, the claim is 'For the inversion pattern, TransE achieves a high value even though it is not expected to jointly capture the pattern.'  TransE was at .09 for inversion compared to say Complex which was at .49.  So how was expectation calculated for each of these embedding methods?
4.  Similarly, when you first introduce partial completeness, it would be good to describe exactly what it is in a bit more detail.  My understanding is that PCA is accomplished by corrupting the subject or object of a triple so that triples with the same say, s and p exist, but o is different.  The text however describes it as: "For instance, triples that fulfill p(x, y) ∧ p(y′, x) where y ≠ y′ are negatives according to PCA[14]" - notice that there is also an inversion here.  Is that how you used it in this paper?
5.  In table 2, why is the negative evidence not summarized?  It is in Table 3 for the new patterns.

**Reviewer Confidence:**

3: The reviewer is confident but not certain that the evaluation is correct

**Scope:**

4: The work is relevant to the Web and to the track, and is of broad interest to the community

---

### Official Review · Reviewer_zJDt · 2023-11-21

**Novelty:** 6
**Technical Quality:** 5

**Review:**

This article offers a methodology and a study to quantify to which extent a KGE model captures different inference patterns expressed as Horn rule "templates". The proposed methodology exploits some key considerations of KGE learning and resorts to the partial completeness assumption (PCA) to generate counter-evidence when training the KGEs. The study includes a varied selection of KGE models, inference patterns, and also studies the KGE's abilities to capture counter-evidence.

Overall the contribution of this article is valuable to the KG research community as it sheds lights on the structural patterns that are preserved by existing KGE techniques -- which are the de facto standard nowadays for multiple inference tasks. The varied selection of KGEs is also well-appreciated.

The paper's related work is quite comprehensive. I would recommend the authors to have a look at this work: https://arxiv.org/abs/2209.08858. It provides a theoretical analysis of how incompleteness affects the utility of standard performance metrics for link prediction.

The methodological procedure to evaluate the preservation of the inference patterns is also interesting, and makes sense. It is based on the observation that embeddings are trained to rank seen facts better than potentially false ones. The authors then exploit this to focus on those triples deemed as likely by the KGE. The authors then compare the support of the inference patterns in that selected set vs. in the entire dataset (to be precise they focus on the test set of the benchmark datasets).

Overall, this article has some merits. I think nevertheless that the paper should also answer the following questions.

1) How many counter-examples per positive triple were used to train the embeddings? Did you use the same in every epoch? This may explain why the models do not capture PCA very well -- they have not seen enough counter-evidence to capture the PCA. Also, the PCA is usually assumed on one direction, e.g., for citizenOf(J. Biden, USA) the PCA will assume that J. Biden does not have any other nationality -- conversely, it is difficult to assume we know all citizens of USA.
Giving more details about the handling of negative examples may shed more light on why the models do not capture negative evidence properly.

2) I am also a bit skeptical of whether focusing only on a confidence threshold of 0.1 makes sense. Patterns with low confidence could be identified as noise by the KGE. For this reason it would be have been nice to see this analysis at different confidence thresholds - if an inference pattern is very obvious, e.g., conf > 0.5 and is not captured by a KGE, then we know for sure the KGE is not good at preserving it.

I know space is limited, but an analysis for k=3 would have been informative as the KGEs may be good for link prediction of some predicates -- specially easy functional predicates.

Finally, I think the paper's takeaways are too shallow, specially given the amount of information presented in Tables 2 and 3 and the significant work behind it (by the way the arrows coding is painfully confusing). Redundancy helps models capture some patterns better. Why is it that? Is it because it makes the graph more densely connected? What do the authors mean with "a pattern being captured both independently and jointly"? What is the meaning of the green and pink color coding? Unfortunately the evaluation and analysis suffers from a few clarity issues that prevent from appreciating all the insights that could be unveiled by this interesting experimental protocol.

**Questions:**

See two questions in the review text.

**Ethics Review Description:**

No issues

**Reviewer Confidence:**

3: The reviewer is confident but not certain that the evaluation is correct

**Scope:**

3: The work is somewhat relevant to the Web and to the track, and is of narrow interest to a sub-community

---

### Official Review · Reviewer_ccFE · 2023-11-26

**Novelty:** 5
**Technical Quality:** 5

**Review:**

# Overall

This work defines a method to assess the capacity of inference patterns captured by embedding models. This is a relevant topic.

Next I present minor issues of the paper.

# Issues

## Section 1

1. The authors say that graphs are typically incomplete *due to their unsupervised construction*. I do not understand what they mean for *unsupervised construction*. In general, databases can be incomplete for many reasons. For example, the data could have been removed for privacy concerns, or it just was not available when the database was created.

3. It is said that $\mathrm{meronym}(w_1, w_2) \to \mathrm{holonym}(w_2, w_1)$ is a symmetry pattern. However, it is not because symmetry is a property of a single relation. In this case, it is an inversion pattern. This issue also occurs in Section 2.1.

## Section 2.1

1. It is said that *other methods that study inference patterns are [...]*. However, methods do not study things. I guess that the authors indented to write that these methods capture patterns in the data. However, I guess that every method should somehow capture patterns. Thus, I do not understand what the authors intended to say. Maybe just mentioning other existing methods.

## Section 2.2

1. It is said that *there is no consistent analysis on how patterns are captured*. I guess that the authors intended to say systematic instead of consistent. Systematic means including all the relevant aspects, whereas consistent means having no contradictions.

2. They mention that *studying predicate cardinality is different from analyzing inference capabilities*. However, according to what they write, it seems that the authors they are referring to are indeed studying inference capabilities regarding predicate cardinalities.

## Section 3

4. This section defines inference patterns to include only two variables. However, Example 3.3 does not follow this definition because the example composition pattern includes three variables.

## Section 4.1

1. Before example 4.2, it must be $\phi(Y) \neq \phi(Y')$ instead of $Y \neq Y'$ because we already know $Y$ and $Y'$ are different variables. This condition should be included in the condition of the definition of the set $\mathrm{neg}(l_i, G)$.

2. Identity (4) differs from the partial completeness assumption described in [14] because identity (4) does not exclude the positive triples from the set $neg(l_i, G)$. Indeed, consider a graph $G = \lbrace r(a,b), r(a,c), p(a,b), p(a,c) \rbrace$ and the rule $r(x, y) \to p(x, y)$. According to [14], $p(a, b)$ and $p(a, c)$ are in $\mathrm{KBtrue}$, and thus not in $\mathrm{KBfalse} \cup \mathrm{NewKBfalse}$. Hence, according to [14], $p(a, b)$ and $p(a, c)$ are not negative. However, according to equation (4), $p(a, b)$ and $p(a, c)$ are negative.

**Questions:**

I wonder to know if the previous patterns and the proposed patterns cover all possible patterns where the body has at most two atoms. If this is not true, what patterns are excluded and why? We can consider that each pattern is an equivalence class of isomorphic rules under renaming variables and predicates.

**Ethics Review Description:**

No ethic flags

**Reviewer Confidence:**

2: The reviewer is willing to defend the evaluation, but it is likely that the reviewer did not understand parts of the paper

**Scope:**

4: The work is relevant to the Web and to the track, and is of broad interest to the community

---

### Official Review · Reviewer_T6eu · 2023-12-01

**Novelty:** 5
**Technical Quality:** 6

**Review:**

The paper can be seen as a significant advancement of “[1] RalphAbboud,İsmailİlkanCeylan,ThomasLukasiewicz,andTommasoSalvatori. 2020. BoxE: A Box Embedding Model for Knowledge Base Completion. In NeurIPS. 9649–9661” in so far that more patterns are included and the capture of negative evidence in terms of learning the partial completeness assumption is concerned.

The paper provides several significant contributions:

* Table 1 can be viewed as a survey regarding the theoretical analysis of pattern inference for different embedding-based models for KG completion. The table clarifies, for example, that the theoretical analysis often comes well after the introduction of the model and that an analysis is not even known for a good amount of model x pattern pairs.
* Section 4 provide a model-agnostic method for determining (quantifying) the capture of patterns including negative evidence (PCA). To this end, a calculus for collecting predicted links, also based on standard support for positive evidence and a customized support function for negative evidence (which takes PCA into account).
* Table 2 is the main result as it effectively shadows known theoretical analysis results with uniform measurement results based on Section 4. We get to see, for example, which model performs the best for what patterns and whether the finding over- or underperforms regarding existing (theoretical?) analysis.
* Table 3 covers additional inference patterns not covered much in previous work, e.g., generic intersection.

**Questions:**

In Table 1, it says “If a theoretical analysis is missing, we report the first reference that did it.” Perhaps, clarify that this is about the original embedding paper (left column) not having performed a theoretical analysis, but instead the added reference is doing it, ok?

Regarding the definition of negative triples, “PCA [7, 11] deems negative triples those that are not present in G when the subject or object, but not both, of a given triple is corrupted”, I have a stupid question. How would we make sure that by introducing negative triples in accordance with this definition that we don’t accidentally create a triple that is in accordance with a pattern that we still have to discover? (Or do we specifically generated negative evidence from assumed patterns, which is however not what they quote says?) It looks like Section 4.1 gets into this.

Section 5.2 has an inline table for AMIE-mined patterns. This could become a properly labeled table. What are the values in the cells? (What’s the scale)

**Reviewer Confidence:**

1: The reviewer's evaluation is an educated guess

**Scope:**

3: The work is somewhat relevant to the Web and to the track, and is of narrow interest to a sub-community

---

### Decision · Program_Chairs · 2024-01-22

**Decision:**

Accept (Oral)

**Comment:**

Overall the reviewers agree this paper tackles a relevant problem for the KG community (preserving the semantics of KGEs). The few concerns about the method and experiments seems to be clarified after rebuttal, and these should be addressed if the paper is accepted. Nonetheless the contribution to the field is significant and I recommend for acceptance .